# Spreading of Aggregated α-Synuclein in Sagittal Organotypic Mouse Brain Slices

**DOI:** 10.3390/biom12020163

**Published:** 2022-01-19

**Authors:** Buket Uçar, Nadia Stefanova, Christian Humpel

**Affiliations:** 1Laboratory of Psychiatry and Experimental Alzheimer’s Research, Department of Psychiatry and Psychotherapy, Medical University of Innsbruck, Anichstrasse 35, A-6020 Innsbruck, Austria; Buket.Ucar@i-med.ac.at; 2Laboratory for Translational Neurodegeneration Research, Division of Neurobiology, Department of Neurology, Medical University of Innsbruck, Innrain 66, A-6020 Innsbruck, Austria; Nadia.Stefanova@i-med.ac.at

**Keywords:** spreading, α-synuclein, organotypic brain slices, pre-formed fibrils, PLP transgenic mice

## Abstract

The accumulation of α-synuclein (α-syn) in the brain plays a role in synucleinopathies and it is hypothesized to spread in a prion-like fashion between connected brain regions. In the present study, we aim to investigate this spreading in well-characterized sagittal organotypic whole brain slices taken from postnatal wild type (WT) and transgenic mice overexpressing human α-syn under the promoter of proteolipid protein (PLP). Collagen hydrogels were loaded with monomers of human α-syn, as well as human and mouse pre-formed fibrils (PFFs), to allow local application and slow release. The spreading of α-syn was evaluated in different brain regions by immunohistochemistry for total α-syn and α-syn phosphorylated at the serine129 position (α-syn-P). The application of human and mouse PFFs of α-syn caused the aggregation and spreading of α-syn-P in the brain slices, which was pronounced the most at the region of hydrogel application and surrounding striatum, as well as along the median forebrain bundle. The organotypic slices from transgenic mice showed significantly more α-syn pathology than those from WT mice. The present study demonstrates that seeding with α-syn PFFs but not monomers induced intracellular α-syn pathology, which was significantly more prominent in brain slices with α-syn overexpression. This is consistent with the prion-like spreading theory of α-syn aggregates. The sagittal whole brain slices characterized in this study carry the potential to be used as a novel model to study α-syn pathology.

## 1. Introduction

Alpha-synuclein (α-syn) is a 140-amino-acid protein that is widely expressed in the brain. The accumulation of insoluble α-syn aggregates is the hallmark pathology of synucleinopathies that include Parkinson’s disease (PD), dementia with Lewy bodies (DLB) and multiple system atrophy (MSA). The physiological function of α-syn is not fully clear, but there is evidence that it is involved in vesicle trafficking and synaptic functions [1]. The aggregates of α-syn are concentrated in neurons, forming large Lewy bodies and Lewy neurites in PD [2]. In MSA, these aggregates accumulate in oligodendrocytes and are termed glial cytoplasmic inclusions (GCIs) [3].

The *prion hypothesis* is based on the idea that neurodegenerative diseases are caused by a systemic aggregation and transport of specific misfolded proteins in the brain through axonal pathways [4]. Increasing evidence shows that some proteins (e.g., beta-amyloid, tau and α-syn) associated with neurodegenerative diseases such as Alzheimer’s disease or PD follow a characteristic spatiotemporal pattern for spreading in the brain [5]. This spreading hypothesis is based on two main mechanisms, namely, the aggregation of the protein by incorporating the native forms into the fibrils and the spreading over distances. Such a mechanism has also been postulated for α-syn in PD. First, it has been well documented that embryonic ventral mesencephalon (vMES) cells transplanted into PD patients displayed Lewy body-like inclusions in post mortem examinations [6,7], suggesting that aggregated α-syn can propagate from diseased cells to healthy ones. Secondly, exogenous pre-formed fibrils (PFFs) of α-syn acted as seeds to recruit endogenous α-syn in vitro [8,9] and in vivo [10,11,12], similar to the self-propagation seen in prions. A single injection of α-syn PFFs into the striatum of wild-type mice induced a Lewy body pathology in the interconnected brain regions accompanied by dopaminergic neuron loss and motor deficits [13], suggesting a relationship between α-syn transmission and PD.

Organotypic brain slices are models that encompass the three-dimensional structure of the in vivo models together with ease of manipulation of in vitro models [14]. Considering the involvement of astrocytes and microglia in α-syn spreading and degradation, brain slices simulate a more physiologically relevant model with the neurons within the glial matrix than the isolated, single-cell-type cultures [15]. It has been demonstrated, in hippocampal slices, that the α-syn aggregates could spread between the neurons of interconnected regions anterogradely upon local microinjection of α-syn PFFs [16]. Furthermore, the application of α-syn PFFs to organotypic brain slices caused a time-dependent increase in α-syn aggregates, which was dependent on endogenous α-syn expression [16,17]. Another study showed that different polymorphs of α-syn PFFs had different seeding abilities in organotypic hippocampal slices [18]. Finally, it was shown that enhanced neuronal activity induced a higher uptake of exogenous α-syn PFFs as well as formation and propagation of α-syn aggregates in hippocampal and midbrain organotypic brain slices [16,17,19]. One study achieved success in culturing resected human brain tissue and inducing α-syn pathology ex vivo by viral expression and α-syn PFFs; however, this technique was limited by the availability of suitable tissue and low survival in the culture [20]. Models of a specific brain region are useful tools to investigate the mechanisms behind the spreading of α-syn, especially from the hippocampus, due to its well-characterized connectivity. However, a model with several brain regions is required to observe possible variations in distinct brain areas, besides including different neuronal populations and axonal pathways.

The aim of the present study was to investigate the spreading of different α-syn proteins in combined brain areas using sagittal whole organotypic brain slices. Further, we used our well-characterized collagen hydrogels to allow slow and localized release of α-syn onto these brain slices. We evaluated the spreading of α-syn in wild-type (WT) mice, but also in a transgenic (TG) MSA model overexpressing human α-syn. Our data provide evidence that seeding with α-syn PFFs but not monomers induced α-syn pathology. This effect was significantly more prominent in brain slices with α-syn overexpression. The sagittal whole brain slice model established and characterized in this study is a valuable tool for studying mechanisms underlying α-syn neurotoxicity and testing future therapeutic strategies that target proteinopathies.

## 2. Materials and Methods

### 2.1. Animals

In this study, we used WT (C57BL/6N) and transgenic (TG) mice overexpressing human α-syn under the promoter of oligodendrocyte-specific proteolipid protein (PLP), which were housed at the Medical University of Innsbruck animal facility and were provided with open access to food and water under 12 h/12 h light–dark cycles. Transgenic α-syn mice are well characterized and have been used by us previously [21]. All animal experiments were approved by the Austrian Ministry of Science and Research (66.011/0055-WF/V/3b/2017 and BMWF-66.011/0120-II/3b/2013) and conformed to the Austrian guidelines on animal welfare and experimentation. Our study using animals (mice) followed ethical guidelines for sacrificing animals and our animal work was in compliance with international and national regulations. All our slice experiments were defined as “organ removal” and were not “animal experiments”.

### 2.2. Preparation of Organotypic Brain Slices

Organotypic brain slices were prepared as described previously [22,23], with some modifications. Briefly, postnatal day 8–10 mouse pups were rapidly decapitated and their brains were dissected. The two hemispheres were separated with a scalpel and glued onto the platform of a water-cooled vibratome (Leica VT1000A). Sagittal slices with 200 μm thickness were cut and placed on membrane inserts (Millipore PICM03050), or onto additional membranes (Merck; HTTP02500) and cultured in 6-well plates (Greiner). In each well, 1 mL of culture medium was added, which contained 50% MEM/HEPES (Gibco), 10% heat-inactivated horse serum (Gibco/Lifetech), 25% Hanks’ solution (Gibco), 2 mM NaHCO_3_ (Merck), 6.5 mg/mL glucose (Merck), 2 mM glutamine (Merck), (pH = 7.2) and, additionally, 10 ng/mL glial cell-line-derived neurotrophic factor (GDNF; Prospec; CYT-243). The medium was changed once per week. After 1 week in culture at 37 °C and 5% CO_2_, collagen hydrogels were placed on the striatum of each slice and the slices were incubated for 8 further weeks to follow up spreading. The slices were then either fixed with 4% paraformaldehyde or collected for Western blotting and qRT-PCR. In order to study dopaminergic neurons, 6 vibrosections were cut per hemisphere and made up one experimental group, while the 6 vibrosections from the other hemisphere served as a second experimental group.

### 2.3. Lactate Dehydrogenase (LDH) Assay

The viability of the brain slices during the culturing period was evaluated with a lactate dehydrogenase (LDH) assay. The culture media were collected every week during media change and stored at −20 °C until analysis. As a positive control, the slices were incubated with 2% Triton overnight. The concentration of LDH released into the media, which correlates with the number of apoptotic and necrotic cells, was quantified in a 96-well format according to the manufacturer’s guide. The absorbance was measured at 450 nm with a microplate ELISA reader (Zenyth 3100 ELISA reader) and the overall survival was calculated as the percentage of the positive control.

### 2.4. Preparation of Collagen Hydrogels and α-Syn Proteins

In order to apply the substances onto the slices, non-toxic, degradable and slow releasing collagen hydrogels were placed on the slices. The collagen hydrogels were prepared using 4S-Star poly(ethylene glycol) succinimidyl succinate (Sigma JKA7006; 4S-StarPEG) as a cross-linker, as described previously [23,24]. Briefly, 2 mg/mL bovine collagen type I (Collagen Solutions) was mixed with the crosslinker 0.15 mM 4S-StarPEG in phosphate-buffered saline (PBS) to the final volume of 180 µL. A load of 10 µL of stock solutions of α-syn monomers or PFFs at a 1 µg/µL concentration was added to 90 µL of this hydrogel solution, while control hydrogels were generated by adding the same volume of PBS as the load (pH 7.2). PFF solutions were thawed and kept at room temperature, while the monomeric α-syn solutions were thawed and kept on ice throughout the preparation of hydrogels. Droplets of 2 µL (200 ng α-syn) from the final solution were pipetted onto sterilized Teflon tapes and incubated at 37 °C for 1 h for gelation. Collagen hydrogels were placed onto the striatum of sagittal brain slices that were pre-cultured for 1 week. The following α-syn proteins were used in this study: recombinant human α-syn aggregate (control) (Abcam; ab218817), recombinant human α-syn monomer (active) (Abcam; ab218818), recombinant human α-syn aggregate (active) (Abcam; ab218819) and recombinant mouse α-syn aggregate (active) (Abcam; ab246002). As a control, empty collagen hydrogels or PBS only were tested.

### 2.5. Immunohistochemistry

Immunostainings were performed as described previously [23]. At the end of the culturing period, the slices were fixed with 4% paraformaldehyde for 3 h and washed 3× with 10 mM phosphate-buffered saline (PBS). The slices were incubated for 30 min in 0.1% Triton-PBS (T-PBS) at room temperature (RT) and washed 3× with PBS. The slices were blocked in 20% horse serum/0.2% bovine serum albumin (BSA)/T-PBS for 30 min at RT and subsequently incubated with primary antibodies against α-syn (1:500; Abcam; Ab131508) and phosphorylated S129 α-syn (α-syn-P; 1:500; Abcam; Ab51253 (EP1536Y)) diluted in T-PBS/0.2% BSA at 4 °C for 48 h. After subsequent washing (3× with PBS), the slices were incubated with Alexa-488 anti-rabbit secondary antibodies for 1 h and counterstained with DAPI. The slices were washed and mounted in Mowiol. For the colocalization experiments, the brain slices were stained with α-syn-P with an Alexa-488 conjugated secondary antibody in addition to antibodies against NeuN (Millipore; MAB377), MAP2 (Abcam; Ab5392) and GFAP (Millipore; AB5541) with their respective Alexa-546-conjugated secondary antibodies. Dopaminergic neurons were visualized by using primary antibody against tyrosine hydroxylase (TH; Proteintech; 25859-1-AP) with Alexa-488 anti-rabbit secondary antibody. For neurotracing, Mini-Ruby crystals (Invitrogen; D3312) were applied onto the striatum of the brain slice with the tip of a needle and the slices were incubated for 2 h. Staining was visualized with a Leica DMIRB inverse microscope and Openlab software (4.0.4). Confocal microscopy was performed using an SP8 confocal microscope (Leica Microsystems, Wetzlar, Germany) with 1.3 NA glycerol objective. Emission of fluorophores were detected from 493 to 556 nm (AlexaFluor 488) and from 566 to 628 nm (AlexaFluor 546). The images were deconvoluted by Huygens Professional software and reconstructed with Imaris 8.2 software.

### 2.6. Release Experiments

The release profile of α-syn from collagen hydrogels was calculated over 8 weeks in an acellular release system and on the organotypic brain slices. Three collagen hydrogels (loaded with human α-syn PFFs or PBS) were placed on small pieces of sterilized parafilm, submerged in 500 µL of culture medium in 24-well plates and incubated at 37 °C and 5% CO_2_ for up to 8 weeks. In an additional experiment, the media were collected from wells containing 3 brain slices with a collagen hydrogel loaded with human α-syn PFFs on each. The media were collected every week and stored at −20 °C until the analyses. The release of α-syn into the media was measured using a human α-syn ELISA kit (RayBio, ELH-SNCA), according to the manufacturer’s protocol. Briefly, standards and samples were pipetted into the wells and incubated for 2.5 h on a shaker; the wells were washed, biotinylated anti-human α-syn antibody was added and the wells incubated for 1 h. After washing away unbound biotinylated antibody, the HRP-conjugated streptavidin (45 min) was pipetted to the wells, incubated and washed; a TMB substrate solution was added to the wells for 30 min, developed and stopped, then, the color was measured at 450 nm. The detection limit was 85 pg/mL.

### 2.7. Western Blotting

Western blotting was performed as described by us [25]. The brain slices were scrapped off the membrane and collected in tubes. The slices were homogenized in 60 µL of PBS with protease inhibitor cocktail (P-8340, Sigma) by sonicating. This solution was centrifuged at 14,000× *g* for 20 min at 4 °C and the supernatant was collected. The total protein amount was detected performing a Bradford assay with Coomassie brilliant blue G250 dye (Bio-Rad). From the organotypic brain slices, 25 µg of total protein (or 50 ng standard) was loaded to the 10% Bis-Tris polyacrylamide gel (Invitrogen) with 2 μL of sample reducing agent and 5 μL of sample buffer. Electrophoresis was performed for 35 min at 200 V. The samples were electrotransferred onto a PVDF membrane for 20 min at 25 V in a semi-dry transfer cell (Thermo Scientific). Blotting was performed using a WesternBreeze Chemiluminescent immunodetection system (Invitrogen). The blots were blocked for 30 min and incubated overnight on a shaker at 4 °C with primary antibodies against neurofilament (1:10,000; Novus; NB300-135), MOG (1:2000; Proteintech; 12690-1-AP), CD11b (1:2000; Proteintech; 20991-1-AP), GFAP (1:1000; Millipore; AB5541), α-syn (1:1000), α-syn-P (1:1000) and actin (1:1000; Sigma; A2066). The blots were incubated with alkaline phosphatase-conjugated secondary antibody (anti-chicken for GFAP and anti-rabbit for others) for 30 min at room temperature. Following a brief washing step, the blots were incubated in a CDP-Star chemiluminescent substrate solution (Roche) for 15 min and visualized with a cooled CCD camera (SearchLight; Thermo Scientific).

### 2.8. Quantitative Real Time Polymerase Chain Reaction (qRT-PCR)

The slices were scraped off from the membrane and collected into tubes. The collected tissue was homogenized by sonication and using a QIA-Shredder kit (Quiagen, Germany). Total RNA was extracted with a RNeasy Mini kit (Qiagen, Germany), according to the manufacturer’s protocol. Total RNA concentrations were determined using Nanoquant Infinite M200 (Tecan, Switzerland). Reverse transcription was performed from 75 ng of total RNA in 20 μL and carried out with a Omniscript Reverse Transcription kit (Quiagen, Germany), including random hexamer primers (Promega) and RNAse inhibitor (Sigma), according to the manufacturer’s protocol. qRT-PCR was performed as described previously [23]. The relative abundance of endogenous α-syn was assessed by TaqMan quantitative PCR using a standard curve method based on normalization to housekeeping gene glyceraldehyde 3-phosphate dehydrogenase (GAPDH). A TaqMan gene expression assay specific for the *Snca* gene designed to span exon–exon boundary (Mm01188700-m1) and a Gapdh assay (Mm99999915-g1) were used (Applied Biosystems). qRT-PCR was performed as 50 cycles (Ct = 20–24) in duplicates, using 3.2 μL of total RNA equivalents of cDNA and the specific TaqMan gene expression assay for each 20 μL of the reaction in TaqMan Universal PCR Master Mix (Applied Biosystems). The analyses were performed using QuantStudio 6 (Applied Biosystems) and the Ct values for each gene expression assay were recorded for each individual reaction. All experiments were normalized to housekeeping *Gapdh* and relative expression was calculated using the ΔΔCt method.

### 2.9. Data Analysis and Statistics

The amount of protein aggregation and spreading was evaluated blindly by calculating the total area stained by α-syn-P antibody. The images were captured from the whole slices at a 60× magnification for slices obtained from WT mice and a 10× magnification for slices taken from transgenic PLP mice due to excessive fluorescent staining in these slices. The images were captured with the same mode and light settings within each group. The images were analyzed by ImageJ by applying a global threshold and converting them into binary images. The area stained by the α-syn-P antibody was calculated for each brain region for each brain slice. The number of dopaminergic neurons was quantified by counting TH+ neurons at vMES blinded under the microscope at a 10× magnification. Neurons were counted if a clear nucleus and/or TH+ extensions were present. The total number of the dopaminergic neurons from one experimental group of 6 slices that were taken from one hemisphere was calculated. All values are given as mean ± standard error of the mean (SEM). Sample size (*n*) always indicates the number of animals. The statistical analyses were performed by a one-way ANOVA with a subsequent Fisher LSD post hoc test and *p* < 0.05 represented significance.

## 3. Results

### 3.1. Viability of Organotypic Brain Slices

All the slices used in the experiments were morphologically evaluated in terms of tissue thinning, adherence to the membrane and change in color and damaged slices were excluded from the study. Sagittal organotypic brain slices were cultured on semipermeable membrane inserts and adhered well within the first week (Figure 1A). After 1 week, collagen hydrogels loaded with or without α-syn proteins were placed on the striatum of each sagittal slice (Figure 1B, blue circle). The brain slices were cultured for 8 further weeks and the aggregation and spreading of α-syn was assessed in five brain regions (Figure 1B,C). Eight weeks after incubation, the brain slices maintained their structure as seen after staining with blue fluorescent nuclear DAPI (Figure 1C). The Western blots showed a stable expression of neuronal neurofilament, astroglial GFAP, oligodendroglial MOG and microglial CD11b eight weeks after treatment with the different α-syn proteins (Figure 1D). Actin always served as a loading control (Figure 1D). The LDH assay correlated with cell damage and tissue survival; it was used to determine cellular viability and did not show any differences between the treatment groups (Figure 1E).

### 3.2. α-Syn Loaded in Collagen Hydrogels

Two days after application of α-syn proteins loaded in collagen hydrogels, a strong green fluorescent (Alexa-488) staining with the α-syn antibody was observed in and around the area of the hydrogel (Figure 2A). At the borders of the hydrogel, some α-syn staining already seemed to be internalized by the cells (Figure 2B). No α-syn staining was visible when a control collagen hydrogel was applied (Figure 2C). After 4 weeks of incubation, most of the collagen hydrogels degraded and aggregates of α-syn-P were found, which seemed to be localized in neurites rather than cell somas (Figure 2D). Eight weeks after culturing, many Lewy body-like round shaped deposits were present, in addition to abundant neuritic aggregates (Figure 2E–G). The specificity of α-syn staining was confirmed as no staining was seen in the red channel (Figure 2H) or by excluding the primary antibody (Figure 2I).

### 3.3. Characterization of α-Syn Proteins by Western Blotting

In order to characterize α-syn proteins, Western blots were used and stained with antibodies against α-syn (Figure 3A) and α-syn-P (Figure 3C). With both antibodies, a 14 kDa α-syn protein was present with all recombinant proteins (control PFFs, active monomer, active human PFFs and active murine PFFs) (Figure 3A,C). In addition, higher molecular bands at 40, 50 and 80 kDa were present for all of the fibrillar proteins, except for the monomeric α-syn (Figure 3A,C). In order to investigate the time-dependent changes in α-syn and α-syn-P, the organotypic brain slices that were treated with human PFF-loaded collagen hydrogels were tested from WT and TG PLP mice. A very weak staining (approx. at 80 kDa) was observed when using the α-syn antibody at all time points, which was slightly darker over time (Figure 3B). However, when using the α-syn-P antibody, a strong signal was seen for 80 kD in WT as well as TG PLP slices, which markedly increased after 4 and 8 weeks of culture (Figure 3D). Actin always served as a loading control (Figure 3E).

### 3.4. Release of α-Syn from Collagen Hydrogels

In order to show the release of α-syn from the slices, ELISAs were used to test the concentration of α-syn in the medium. In a first experiment, α-syn proteins loaded in collagen hydrogels were applied onto parafilms and incubated for 8 weeks. While no α-syn was detectable in empty collagen hydrogels, the release of α-syn loaded to collagen hydrogels reached a maximum already after 3 weeks (Figure 4). In the next experiment, we aimed to show the release of α-syn from slices; however, our data provide evidence that approximately 10% of the loaded α-syn was released into the medium within 8 weeks of incubation, which was largely seen in the first week (Figure 4). This was true for slices taken from WT as well as TG PLP mice (Figure 4).

### 3.5. qRT-PCR of Endogenous α-Syn Expression

In order to study if the application of exogenous α-syn proteins affects the endogenous expression of murine α-syn, a qRT-PCR was performed. Our data show that none of the treatments caused a significant change in the mRNA expression of endogenous murine α-syn in WT slices 8 weeks after culturing (Figure 5) (PBS vs. ColH(-), *p* = 0.302; ColH(-) vs. ColH(hsyn_mon_), *p* = 0.3856; ColH(-) vs. ColH(hsyn_agg(a)_), *p* = 0.1581).

### 3.6. Spreading of α-Syn in Slices Taken from Wild-Type Mice

In order to demonstrate the spreading of α-syn from sagittal WT brain slices, collagen hydrogels loaded with or without aggregated active human α-syn (hSyn_aggr(a)_) were placed on the striatum and the spreading of α-syn pathology was observed 8 weeks after culturing. In brain slices taken and cultured from WT mice, a high endogenous α-syn expression was seen in all investigated brain areas after the application of an empty control collagen hydrogel (Figure 6A,E,I,M). There was no notable difference in α-syn staining in any of the brain regions when the collagen hydrogel loaded with hSyn_aggr(a)_ was applied (Figure 6B,F,J,N). However, there was a marked difference when the slices were stained with the α-syn-P antibody. While no staining was visible in slices treated with an empty collagen hydrogel (Figure 6C,G,K,O), many small α-syn-P aggregates were seen after the application of the hSyn_aggr(a)_ in all investigated brain areas—more pronounced in the area of the hydrogel (Figure 6D,H,L,P).

### 3.7. Spreading of α-Syn in Slices Taken from Transgenic PLP Mice

In order to characterize the untreated PLP slices in terms of α-syn and α-syn-P immunoreactivity, immunostainings were performed from 10-day- and 3-month-old PLP mice, compared to slices cultured for 8 weeks (see Appendix A). A strong α-syn and phosphorylated α-syn immunoreactivity was seen in brains taken from 10-day-old mice (Appendix A), which markedly decreased in 3-month-old mice (Appendix A). In 8-week cultured (P10) slices, a strong α-syn staining was seen (Appendix A), but nearly no α-syn-P staining (Appendix A).

In order to demonstrate the spreading of α-syn from sagittal TG PLP brain slices, collagen hydrogels loaded with or without aggregated active human α-syn (hSyn_aggr(a)_) were placed on the striatum and the spreading of α-syn pathology was observed 8 weeks after culturing. In brain slices taken and cultured from TG PLP mice, a moderate endogenous α-syn expression was seen in all investigated brain areas after the application of an empty control collagen hydrogel (Figure 7A,E,I,M). There was a dramatic increase in α-syn staining in all the brain regions when the collagen hydrogel loaded hSyn_aggr(a)_ had been applied (Figure 7B,F,J,N). However, there was a marked difference when the slices were stained with the α-syn-P antibody. While no staining was visible in slices with an empty collagen hydrogel (Figure 7C,G,K,O), a dramatic increase in the amounts of α-syn-P aggregates was seen after the application of the hSyn_aggr(a)_ in all investigated brain areas—more pronounced in the area of the hydrogel (Figure 7D,H,L,P).

### 3.8. Quantification of α-Syn Aggregation

In order to evaluate α-syn aggregation and spread, the area stained for α-syn-P was quantified. Figure 8A shows a scheme of the α-syn-P pathology in the WT sagittal brain areas. The quantitative analysis (Figure 8B) shows that α-syn-P aggregates were significantly more prominent in the olfactory bulb, in the area of the hydrogel and in the medial forebrain bundle after the application of mouse α-syn_aggr(a)_. Further, the application of human α-syn_aggr(a)_ significantly increased the α-syn-P staining in the area of the hydrogel and in the hippocampus (Figure 8B). No staining was seen after application of the control PBS and empty collagen hydrogel, or the monomeric α-syn (Figure 8B).

Figure 8C shows a scheme of the α-syn-P pathology in the TG PLP sagittal brain areas. In general, α-syn immunoreactivity was dramatically (approx. 1000×) higher in slices from TG PLP mice than in slices taken from WT mice (Figure 8D). The quantitative analysis (Figure 8D) showed that the area stained for α-syn-P was significantly greater in all investigated brain areas after the application of mouse α-syn_aggr(a)_. The application of human α-syn_aggr(a)_ significantly increased α-syn-P in the area of the hydrogel, the medial forebrain bundle, the hippocampus and the ventral mesencephalon (Figure 8D). Further, the application of human aggregated control α-syn significantly increased α-syn-P staining in the area of the hydrogel and the medial forebrain bundle (Figure 8D). No staining was seen after application of control PBS, empty collagen hydrogel or monomeric α-syn (Figure 8D).

### 3.9. Localization of the α-Syn Aggregates

We investigated the cellular localization of the α-syn aggregates with colocalization studies. Phosphorylated α-syn staining had long fibrillary shapes, which seemed to be localized in neuronal extensions (Figure 9A,B). Other aggregates displayed a discontinued line with dot-shaped staining, which may indicate localization in varicosities of neuronal extensions (Figure 9C,D). Aggregates with small fibrillary appearance were observed in the proximity to the nuclei of neurons (Figure 9E and Figure 10A–C). In some neurons, α-syn aggregates noticeably surrounded the neuronal nuclei (Figure 10D–G). Figure 10G shows a nucleus wrapped into the α-syn aggregates and a degenerated axon with high α-syn load. Some α-syn aggregates co-localized with MAP2 are seen in 3D reconstructed images (Figure 10H–J). No co-localization was seen with astroglial GFAP or microglial Iba1 (data not shown).

### 3.10. Effect of α-Syn Aggregates on Dopaminergic Neuron Survival

In order to demonstrate an intact pathway between the striatum and vMES, Mini-Ruby retrograde tracing was performed. Our data clearly show that Mini-Ruby was transferred some 100–300 µm towards the vMES but did not reach the vMES, indicating destruction of the pathway (Figure 11A). Using immunostainings, many TH+ neurons (118 ± 5; *n* = 5) were seen in the vMES after 9 weeks of incubation (Figure 11B,C). These TH+ neurons had a clear nucleus and extensive fibers were detected (Figure 11B). The application of human α-syn PFFs did not (*p* = 0.378) affect the number of dopaminergic neurons (111 ± 16; *n* = 6) compared to the control (Figure 11C) after 8 weeks of incubation with GDNF.

## 4. Discussion

In the present study, we investigate the spreading of exogenously applied α-syn proteins on whole brain sagittal mouse organotypic brain slices. In order to provide a local and concentrated application, we used well-characterized collagen hydrogels loaded with α-syn proteins. We provide a model of synucleinopathy showing that especially the PFFs of α-syn could induce a pathology and this effect was stronger in brain slices taken from transgenic PLP mice that overexpressed α-syn.

(a)
*Organotypic brain slices to study α-syn spreading*


Organotypic brain slice cultures present an intact cytoarchitecture and maintain the anatomical organization of the brain tissue including all cells of the brain [14]. Such a model is an excellent tool to study different aspects of neurodegeneration but also to study neurodegenerative events after the addition of different exogenous proteins [26]. In a very recent study, we studied the spreading of exogenous human beta-amyloid(42) in such organotypic brain slices [25,27]. For investigating the spreading of α-syn, organotypic hippocampal, cerebellar or cortex slice cultures have been previously used [16,17,19]. Although single brain areas are useful to study the local distribution and impact of the α-syn aggregates, single brain region slice models lack the interactions of the brain regions with each other. Therefore, our sagittal whole brain organotypic brain slices had a strong advantage over other models in this respect. They could be easily cultured and have been well characterized by our group [22]. In the present study, we took advantage of the sagittal whole brain slices to explore α-syn spreading and we applied α-syn proteins directly onto the striatum. In fact, the striatum is a receptive region for the application of synthetic α-syn PFFs and it is a connected structure of the nigrostriatal dopaminergic system, which is affected in PD or MSA [12,13].

The viability of the slices was very important, especially as we cultured the brain slices for up to 9 weeks. The reason for long culture times is that the spreading of aggregates is not a rapid process. This was also based on our previous study [25], where spreading of beta-amyloid was tested in brain slices 8 weeks after application. We also decided to apply the exogenous proteins 1 week after setting up the brain slice cultures to provide a recovery period from the trauma caused by tissue slicing. It is well established that healthy brain slices adhere to the membrane, flatten and become transparent within 1 week. This is a sign of good viability; therefore, we excluded slices which did not show these morphological signs. In order to verify the viability of the brain slices, especially after the application of the collagen hydrogels and α-syn proteins, we used Western blotting and an LDH assay. In all groups, there was similar expression of neurofilament, astroglial GFAP and oligodendroglial MOG, as well as low expression of microglial CD11b. Astroglial GFAP staining revealed fragmented proteins as seen in multiple bands on the Western blot, which is usual for organotypic brain slices due to the activation of astrocytes [25]. The overall evaluation of these markers indicated good viability of the slices with no noticeable difference caused by the application of the hydrogel and α-syn PFFs. Furthermore, the LDH assay showed that the application of the α-syn PFFs in collagen hydrogels did not affect the survival of the slices. Overall, our data clearly show that neither application of the collagen hydrogels nor α-syn PFFs caused cell death.

(b)
*Collagen hydrogels for application of α-syn PFFs*


In the present study, we used well-established collagen hydrogels to apply exogenous proteins locally and at high concentrations. Collagen is a natural biomaterial with a variety of applications for drug, cell or gene delivery to the brain [28]. We showed that collagen hydrogels are not cytotoxic and can provide neuroprotection, neurite growth and vascular repair by slow and local delivery of trophic factors and angiogenic factors [24,27]. Furthermore, collagen hydrogels were used to study the spreading of beta-amyloid peptides in organotypic brain slices in our recent study [25]. In fact, the application of α-syn proteins locally in large concentrations constitutes a challenge. Some studies have used simple approaches such as applying the α-syn solution directly onto the slices [17] or into the medium. These methods are not suitable for the local application of proteins, as α-syn is rapidly distributed to all brain regions in contact. Furthermore, the amount of α-syn that actually reaches the tissue is not controllable. More advanced techniques such as microinjection require special equipment [16]. Thus, the use of collagen hydrogels loaded with α-syn is very effective, straight forward and cost efficient. We reported that collagen hydrogels provided a degradation-based release of the loaded proteins and completely degrade after 2 weeks upon placing on organotypic brain slices [24]. We also provide data that collagen hydrogels displayed a good release profile for the local application of α-syn. We show that α-syn was completely released into the medium when loaded in collagen on a parafilm in an acellular release system. However, when α-syn was loaded in collagen hydrogels placed onto the brain slices, only small amounts of α-syn were detected in the medium. This clearly suggests that the applied α-syn is taken up by cells or is degraded.

(c)
*α-syn proteins tested in this study*


In the present study, we tested four different α-syn proteins, human α-syn monomeric-active, human α-syn aggregate-control, human α-syn aggregate-active and mouse α-syn aggregate-active. All proteins were purchased from Abcam and were characterized by the company. In the Western blot, all proteins had a band of 14 kDa; additionally, the PFFs showed higher molecular weight species. In previous studies, exogenous α-syn monomers did not induce α-syn pathology in vitro and in vivo [13,17,29], which is in line with our results. Furthermore, our data show that mouse α-syn PFFs were slightly more potent than human α-syn PFFs in promoting the pathology in the slices. Human and mouse α-syn differ by 7 amino acids in their sequence. It was shown that mouse α-syn PFFs can more efficiently induce fibrillization of soluble α-syn in mouse models than human α-syn PFFs [30]. The larger sequence homology of exogenous α-syn to the endogenous α-syn caused a more efficient seeding in vivo [31]. Another study suggested that mouse α-syn more rapidly formed amyloid fibril structures, possibly due to a lower negative charge and polarity at the C-terminal region of the protein [32]. Therefore, it was expected to see mouse PFFs exerting a more profound spreading effect than human PFFs, although the distribution to the brain areas was similar. In the present study, we also applied human α-syn PFFs in active and control forms. The control and active human PFFs were polymorphs, identical in sequence but different in their fibril structure due to differences in their preparation. While active PFFs had a hydrophobic interface exposed to the outside environment, control aggregates lacked this interface, which restrained them from incorporating endogenous α-syn monomers. In fact, polymorphs were shown to be different in terms of structure, conformation, fibril diameter, nucleation capacity and solubility [4,33]. Synuclein fibril polymorphs were generated based on varieties in solution and affected by pH, ionic strength and number of poly-anions [33]. Active forms of human α-syn PFFs were more potent to induce spreading than control PFFs. Control PFFs were more potent in the transgenic slices, since they overexpressed human α-syn.

(d)
*Characterization of α-syn immunostainings*


In the present study, we used two commercial antibodies, both well characterized by Abcam. The α-syn antibody (ab131508) corresponded to α-syn aa 134–138 and detected total α-syn. The α-syn-P antibody (ab51253) corresponded to human α-syn at the C-terminus (within aa100) and it detected only α-syn phosphorylated at Ser129. Both antibodies recognized all four of the proteins used in Western blotting. The α-syn antibody stained many structures in the brains of WT and transgenic mice probably positive for endogenous mouse α-syn. However, α-syn-P antibody did not detect any structures in the negative control groups from WT and TG mice, but was specific to label exogenous α-syn proteins and formed aggregates. Therefore, it was highly suitable to detect the spreading of the exogenous α-syn PFFs, as well as the phosphorylated aggregates formed via a combination of exogenous and endogenous α-syn. In fact, it is well known that pathological aggregates in both PD and MSA included extensively phosphorylated α-syn at Serine 129 position and phospho-Ser129 α-syn has been used as an indicator of protein aggregation and formation of pathological forms [34]. The phosphorylation of a-synuclein under physiological conditions prepares the protein for degradation [35]. However, it has been proposed that a-synuclein phosphorylation plays an important role in the formation of toxic forms of the protein and hyperphosphorylation is considered a biomarker for a-synucleinopathies [36]. Recent observations suggest that phosphorylation may perturb the interconversions between the oligomeric and fibrillar forms of a-synuclein, thereafter variably interfering with the neuronal homeostasis and the protein’s propagation [37].

(e)
*Spreading of α-syn in slices taken from wild type mice*


In the present study, we used slices taken from postnatal day 8–10 C57BL/6 mice. As discussed previously, we had to use young postnatal mice in order to generate organotypic brain slices. Using WT mice, we wanted to demonstrate the spreading of the exogenously applied α-syn proteins. As mentioned earlier, WT mice had very strong immunostaining for endogenous α-syn. We did not observe any noticeable differences between control groups and slices with recombinant α-syn treatment with the total α-syn antibody, as it was not possible to differentiate the endogenous and the exogenous α-syn. However, using the α-syn-P antibody against phospho-Ser129 α-syn, we succeeded in demonstrating the aggregation and spreading of α-syn proteins. No staining was seen after the application of PBS, an empty collagen hydrogel, monomeric α-syn or control aggregated α-syn protein. However, active aggregated human and mouse α-syn PFFs induced a very strong α-syn pathology around the region of the hydrogel, while small amounts were detected along MFB, hippocampus and vMES. These results clearly indicate the spreading of α-syn aggregates from the striatum towards other brain regions in WT sagittal slices. In Western blotting, aggregation was mainly observed as a large high molecular 80 kDa band. An increase over time in band intensity implicated the recruitment of endogenous α-syn into phosphorylated aggregates; therefore, this supported the prion theory. As the release of α-syn into the medium was very low, it was assumed that some α-syn protein was internalized into cells but that a majority was degraded over time. More work is necessary to determine the enzymes involved in such a degradation process. In order to show if endogenous α-syn expression could be modulated by exogenous α-syn proteins, a qRT-PCR was performed; however, this did not point to any up- or down-regulation of endogenous α-syn expression.

(f)
*Spreading of α-syn in TG PLP organotypic brain slices*


In the present study, we used transgenic PLP mice expressing human α-syn under the oligodendroglial proteolipid protein promoter, which causes the formation of phosphorylated aggregates resembling GCIs [21,38]. The reason to use transgenic mice was to observe if the overexpression of endogenous α-syn may potentiate the aggregation and spreading process. In fact, a previous study showed that treatment with human PFFs caused α-syn-P accumulation in organotypic cerebellar slices taken from two different transgenic mice overexpressing human α-syn [17]. In order to characterize the immunostaining with our two antibodies, we stained fixed sections taken from P8–10-day- and 3-month-old TG mice. We found strong α-syn immunoreactivity in the freshly fixed slices from young mice, which was markedly decreased in older 3-month-old mice. Using our antibody for total α-syn, we found a moderate staining of α-syn in slices cultured for 2 weeks, which may have been due to the specificity of the antibody for mouse but not human synuclein. Interestingly, the addition of exogenous α-syn markedly enhanced the total α-syn immunoreactivity, which was different from WT mice. Using our antibody for phosphorylated α-syn, surprisingly, we did not detect any depositions in control slices cultured for 8 weeks. Similar to the WT groups, no staining was seen after the application of PBS, an empty collagen hydrogel, or monomeric α-syn. It is very likely that some degradation of proteins may have occurred in vitro during the 8 weeks of culture, which is also in line with data indicating that α-syn staining was decreased in 3-month-old mice. However, in the groups treated with exogenous α-syn, a dramatic increase in phosphorylated α-syn immunoreactivity was seen in all areas investigated. This is also in line with another in vitro study showing that cultured oligodendrocytes isolated from PLP transgenic mice did not spontaneously produce aggregated α-syn-P, but increased depositions after treatment with exogenous PFF seeds [39]. α-syn-P staining was the most intensive at the location of the collagen hydrogel, around the striatum and along MFB, as well as in vMES in lower amounts. This α-syn-P staining was much more excessive (approx. 1000×) than that of WT slices with α-syn PFFs. In Western blotting, α-syn aggregation was demonstrated with a high molecular 80 kDa band, which was markedly increased after 4 and 8 weeks in culture. As the release of α-syn into the medium was very low, it is assumed that some α-syn protein degraded over time but that the majority of the exogenous α-syn was internalized and formed intracellular depositions. Most of the aggregates were seen along the medial forebrain bundle as well as in the mesencephalon, pointing to be playing some role in the degeneration of dopaminergic neurons. As mentioned, the intensity in TG mice was more than 1000× higher than that in WT mice; it is assumed that exogenous α-syn potentiated the pathology in TG mice much more due to the overexpression of endogenous protein, which is in line with the prion hypothesis.

(g)
*Cellular localization of α-syn aggregates*


In previous studies on organotypic hippocampal slices, α-syn aggregates were located in neurons, which was shown by colocalization with neurofilament and NeuN staining [16]. We observed aggregates with similar shapes, such as short and long serpentine structures, that were likely to be localized in axons, as well as round-shaped aggregates in cell bodies. With confocal imaging, we observed a partial colocalization of α-syn aggregates with neuronal marker MAP2. We also observed thread-like cellular inclusions surrounding the nuclei, similar to a previous study [16]. We show that these aggregates were located in MAP2+ neuronal processes. Additionally, some smaller, dot-like and elongated aggregates were located in MAP2+ axons and dendrites. However, longer fibrils did not clearly co-localize with the neuronal marker, which may indicate that some fibers with excessive α-syn accumulation degenerated or lost their neuronal markers. In previous studies, it was shown that α-syn could interact with membranes such as cell membrane, mitochondrial membranes and nuclear envelope [40,41]. This interaction caused the disruption of membranes, eventually causing degeneration [42]. Therefore, α-syn fibrils surrounding the nucleus could be one of the fundamental steps leading to the degeneration of the neuron.

(h)
*Effect of α-syn aggregation on dopaminergic neurons*


In the present study, we used sagittal brain slices, which allowed us to investigate dopaminergic neurons in the ventral mesencephalon. In order to catch all the dopaminergic neurons, a consecutive series of 200 µm brain slices was sectioned from one mouse and cultured. In fact, many dopaminergic neurons survived culturing for 9 weeks when incubated with GDNF. In a previous study [22], we already reported that the striato-nigral pathway was cut during vibrosectioning. This was not surprising, as the pathway was angular and was not completely intact in the 200 µm slice. In order to verify this finding, we applied Mini-Ruby on the striatum and did not see a continuous transport to the vMES. Mini-Ruby is a fluorescently tagged biotinylated dextran amine and it is rapidly taken up by neurons and transported anterogradely and retrogradely; further, it labels connected cellular pathways. This is important to show that we can exclude that α-syn is taken up in the striatum and transported via the nigrostriatal pathway. Thus, our data suggest that α-syn must reach the vMES via another mechanism and not the nigrostriatal pathway. The transfer of α-syn can occur via newly formed synaptic connections during culturing, tunneling nanotubules between neurons, secretion of aggregates, or other mechanisms [43]. In the next step, we wanted to explore if α-syn could affect the dopaminergic neurons, as we showed that α-syn spread to the vMES. In vivo models showed that intrastriatal injection of α-syn PFFs caused the formation of Lewy body-like inclusions and subsequent specific degeneration of the dopaminergic neurons in the SN [13]. Similarly, in PLP transgenic mouse, α-syn oligomers are generated at a later stage of life in vivo, i.e., 6 months of age, at the time of nigral neurodegeneration [21]. Therefore, it is considered that α-syn oligomers play a pivotal role in the neurodegeneration in vivo. In organotypic brain slices, the application of α-syn PFFs caused significant neurotoxicity after 4 weeks in culture, correlating with α-syn aggregation [17]. Furthermore, it was shown in a previous study [19] with organotypic vMES slice cultures, that the application of α-syn PFFs caused a slight decrease in the number of TH+ neurons, which was enhanced significantly by increased neuronal activity. In our model, we demonstrated that the application of α-syn PFFs did not affect the number of dopaminergic neurons. This suggests that the amount of α-syn load was too low in the vMES, that exogenous GDNF in the medium protected dopaminergic neurons, or that other cellular processes counteracted α-syn toxicity.

(i)
*Mechanism of α-syn aggregation and spreading*


The spreading of pathological proteins is a complex and long-lasting process. It is suggested that abnormal proteins such as α-syn in PD/MSA and beta-amyloid or tau in AD move from neuron to neuron. This can occur via chemical signals, inflammation, excitotoxicity, loss of trophic support or direct neuronal transfer (prion-like) [44]. It is not fully clear how infectious protein move from one neuron to another. Three hypothetical routes may be involved, i.e., (1) via secreted small vesicles, the exosomes; (2) via thin membranous tunnel nanotubes connecting neurons; or (3) via release and uptake of naked infectious protein. All these must involve moving over long distances (see review in [44]). The process of “infection” may involve the non-classical release of infectious protein (e.g., α-syn, beta-amyloid or tau) into the extracellular space; aggregation, which may also occur before release; travel in the interstitial fluid; uptake by selective vulnerable neurons or glial cells; induction of a misfold in the host cell; or repeat and potentiation of this step. The process of α-syn spreading is not fully clear. Definitely, we can exclude the diffusion of α-syn via the medium, as not much was released. The exogenous α-syn seemed to be either degraded by yet unknown enzymes, or taken up by cell and seed aggregation of endogenous α-syn. In a previous study, we showed that beta-amyloid was taken up by microglia [25]. Although we could not determine a microglial pattern and co-localization, we do not suggest a microglial uptake. Due to the distribution and form of deposition, we rather suggest that it was a neuronal staining, as also some depositions were located at the medial forebrain bundle and in the ventral mesencephalon, where the dopaminergic neurons are located. As a second mechanism, we suggest the potentiation of α-syn pathology, at least in the transgenic mouse model. There was a dramatic extensive increase in α-syn depositions in TG mice compared to WT mice. It seems likely that, in TG mice, (a) most of the exogenously added α-syn formed cellular depositions, (b) that less of the exogenous α-syn was degraded, or (c) that the exogenous α-syn activated endogenous α-syn in the TG model. There was more endogenous α-syn which provided more substrate for seeding for the same amount of exogenous PFFs. It is very likely that exogenous added α-syn indeed activated the α-syn system in TG mice to enhance the formation of α-syn aggregates. This is in line with several other publications using transgenic mice, where seeds can activate a pathological cascade.

(j)
*Limitations of the Study*


This study has advantages but also some limitations. There are various in vitro and in vivo models for the investigation of α-syn aggregation and spreading and the neurotoxic effects of the pathology (for reviews, see [45,46]). Despite their usefulness and practicality, in vitro models are too simplistic to study the spreading of α-syn, as they lack the cellular diversity of the brain tissue. There are various well-established in vivo models; however, these are time consuming and labor intensive, as well as unable to allow one to perform direct manipulation and observation. Organotypic brain slices constitute a favorable alternative by bringing together practicality and conserved tissue architecture (for a review, see [14]).

An advantage of this study is that we could incubate our slices for a long time period, over 9 weeks, which is essential to study spreading processes. This is an advantage over similar studies with organotypic brain slices after treatment with α-syn PFFs [16,17]. In our hands, organotypic brain slices showed good viability and α-syn was able to spread among the brain regions. The possibility of conducting a study for longer time periods makes this model ideal for long-term studies, especially for drug testing. A technical limitation of the study was that α-syn PFFs were not sonicated before application due to the very small volumes used in the experiments. The sonication of PFFs forms smaller fragments and oligomers, which increases the number of seeds. In fact, sonicated α-syn PFFs were the most efficient in potentiating phosphorylated aggregates, compared to non-sonicated, larger PFFs [47]. On the other hand, the use of collagen hydrogels to apply exogenous proteins is outstanding and allows one to perform a local application of high amounts of protein. A major limitation using postnatal organotypic brain slices is that they do not represent an adult brain, which is most likely a key mechanism in PD or MSA. Although we can principally culture organotypic brain slice from adult transgenic mice [48], neuronal viability in such adult slices is reduced. However, this is not necessarily a disadvantage, as others showed that organotypic brain slice cultures from postnatal transgenic mice displayed a pathology similar to Alzheimer’s disease.

## 5. Conclusions

In this study, we succeeded in establishing a sagittal whole brain organotypic brain slice culture model to investigate the aggregation and spreading of α-syn. Our results, in this study, are consistent with the prion-like spreading theory of α-syn aggregates. Only active human or mouse PFFs were able to induce α-syn-P aggregation in WT brain slices. Organotypic brain slices from α-syn overexpressing PLP transgenic mice showed boosted immunoreactivity when exogenous α-syn species (monomer and PFFs) were applied. The model characterized here had the advantage of including different brain areas and neuronal populations compared to previous organotypic brain slice models with a single brain region. Sagittal organotypic brain slice model can be used in future research on the neurotoxic mechanisms of α-syn, as well as therapeutic approaches and drug screening.

## Figures and Tables

**Figure 1 biomolecules-12-00163-f001:**
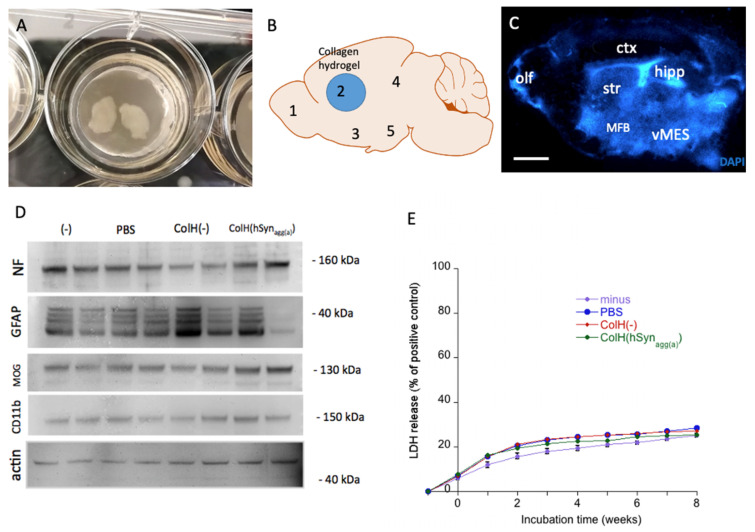
*Set up of the organotypic brain slices and viability.* Sagittal organotypic whole brain slices were cultured on semipermeable membrane inserts (**A**). After 1 week in culture, a collagen hydrogel was placed on the striatum (blue circle) and the spread of α-syn was quantified in 5 brain areas (**B**). The collagen hydrogel (ColH) was loaded without protein (ColH(-)) or with active human aggregated α-syn (CollH(hSyn_agg(a_)). As a control nothing (-) or phosphate-buffered saline (PBS) was added to the slice. After 8 additional weeks in culture, the brain sections were flattened and had an intact morphology as seen with blue nuclear DAPI staining (**C**). The viability of the slices after the different treatments was observed with Western blotting for neurofilament (NF), glial fibrillary acidic protein (GFAP), myelin oligodendrocyte glycoprotein (MOG) and microglial CD11b, where actin was used as a loading control (**D**). Lactate dehydrogenase (LDH) assay was used for measuring viability after 8 weeks in culture. LDH activity was calculated as the percentage of a positive control treated with 2% Triton overnight (**E**). The statistical analyses were performed by one-way ANOVA with a subsequent Fisher LSD post hoc test. Abbreviations: olf, olfactory; str, striatum; ctx, cortex, hipp, hippocampus; vMES, ventral mesencephalon; MFB, medial forebrain bundle. Scale bar, C = 6000 µm (**A**) and 960 µm (**C**).

**Figure 2 biomolecules-12-00163-f002:**
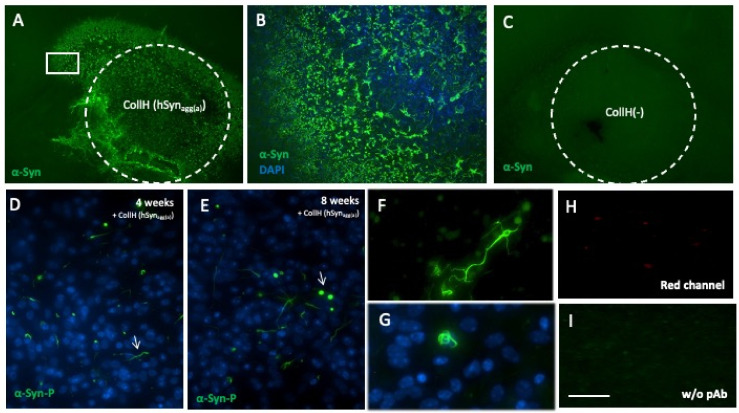
*Characterization of alphα-synuclein (α-syn)-like immunoreactivity loaded in collagen hydrogels to slices.* Collagen hydrogels (white circle in **A**,**C**) loaded without (**C**) or with active human aggregated α-syn (CollH(hSyn_agg(a_); **A**,**B**,**D**–**I**) were applied onto the slices and analyzed after 2 days (**A**–**C**), 4 weeks (**D**) or 8 weeks (**E**–**I**). Two days after incubation, a strong green fluorescent (Alexa-488) immunostaining for α-syn was found on slices where α-syn protein was loaded (**A**,**B**) but not on slices without protein (**C**). (**B**) is a magnification of (**A**) (white square). Immunostainings for α-syn (α-syn-P) show some intracellular depositions and aggregation and Lewy body-like round fibrils after 4 (**D**) and 8 weeks (**E**–**G**) of incubation. No staining was seen in the red channel (**H**) and no staining was seen by omitting the primary antibody (**I**) showing the specific staining. Scale bar, I = 470 µm (**A**), 95 µm (**B**), 470 µm (**C**), 300 µm (**D**,**E**) and 25 µm (**F**–**I**).

**Figure 3 biomolecules-12-00163-f003:**
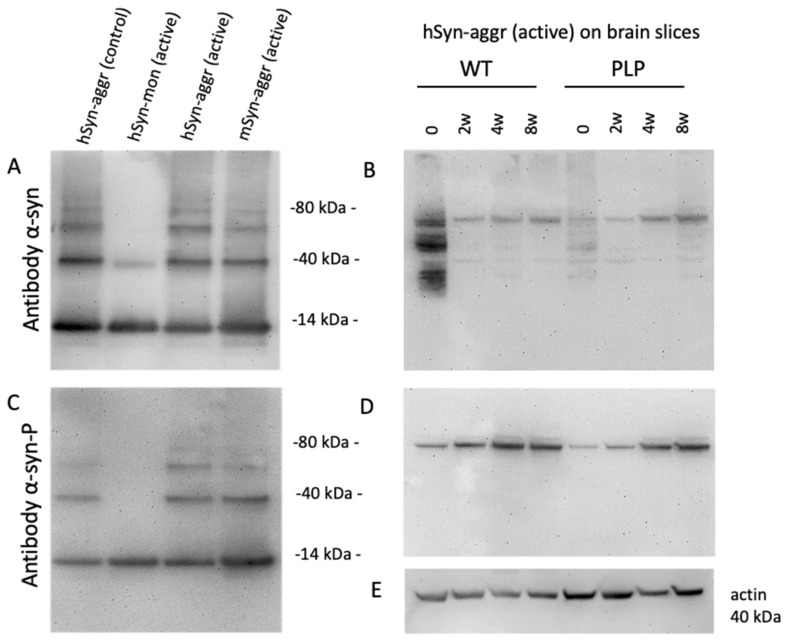
*Characterization of alphα-synuclein (α-syn) proteins by Western blotting and time-dependent changes during culturing.* Four different α-syn proteins were separated on gels, i.e., human α-syn aggregate (control), human α-syn monomer (active), human α-syn aggregate (active) and mouse α-syn aggregate (active). Blots were stained for α-syn (**A**) or phosphorylated α-syn (**C**). All proteins showed a strong band at 14 kDa and some higher molecular weight bands (40 and 60 kDa) were visible for the α-syn aggregates but not for the monomeric α-syn (**A**). Similar results were seen when the gel was stained for α-syn-P (**C**). Slices from WT and transgenic PLP mice were incubated for 0, 2, 4 and 8 weeks with collagen hydrogels loaded with aggregated active human α-syn and stained for α-syn (**B**) and α-syn-P (**D**). Note a strong increase in phosphorylated α-syn-P after 4 and 8 weeks in culture (**D**). Actin was used as a loading control, showing a 40 kDa band (**E**).

**Figure 4 biomolecules-12-00163-f004:**
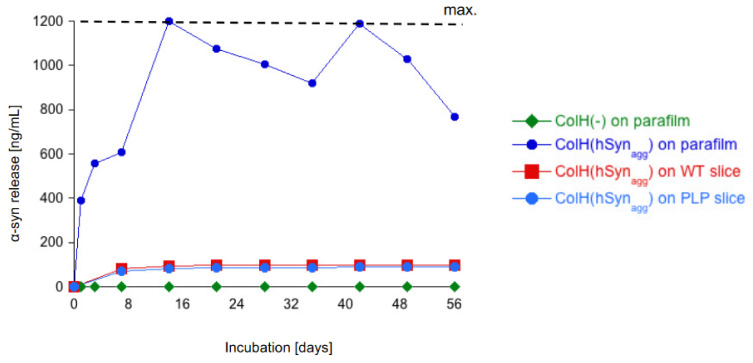
*Release of human alphα-synuclein (α-syn) from collagen hydrogels.* Collagen hydrogels (ColH) were loaded without (-) or with human α-syn aggregate (active) (hSyn_agg(a_), placed on a parafilm and incubated in culture medium for up to 8 weeks. The media were collected every week. In order to evaluate the release from slices, collagen hydrogels loaded with human α-syn_agg(a)_ were applied onto wild-type (WT) and transgenic PLP brain slices. The media were collected each week. The release of α-syn into culture media was measured with a human-specific α-syn ELISA. Values are given as ng/mL. Note that the maximal release was 1200 ng/mL, which was seen when hSyn_agg(a)_ was applied onto a parafilm after 3 weeks. The release of hSyn_agg(a)_ from brain slices into media was very low.

**Figure 5 biomolecules-12-00163-f005:**
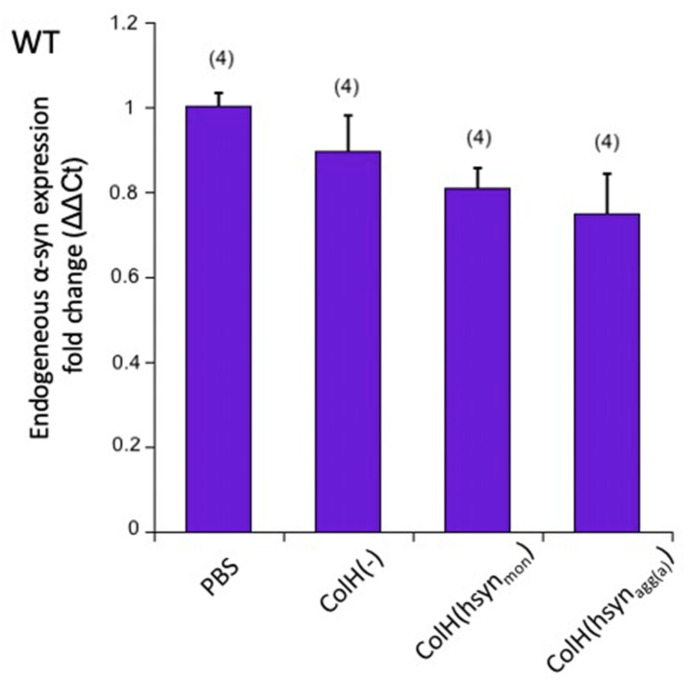
*qRT-PCR for endogenous mouse alphα-synuclein (α-syn).* Collagen hydrogels were loaded without (ColH(-)) or with monomeric (ColH(hsyn_mon_)) or aggregated (ColH(hsyn_agg(a)_)) α-syn and then applied on wild-type (WT) brain slices and incubated for 8 weeks. As a control, only phosphate-buffered saline (PBS) was added. Brain slices were scrapped off the membrane, RNA was isolated and qRT-PCR was performed for endogenous mouse α-syn. The expression of α-syn was compared against the housekeeping gene *gapdh* (*n* = 4 mice). Values are given as mean ± SEM. The statistical analyses were performed by a one-way ANOVA with a subsequent Fisher LSD post hoc test. None of the treatments caused significant differences (PBS vs. ColH(-), *p* = 0.302; ColH(-) vs. ColH(hsyn_mon_), *p* = 0.3856; ColH(-) vs. ColH(hsyn_agg(a)_), *p* = 0.1581).

**Figure 6 biomolecules-12-00163-f006:**
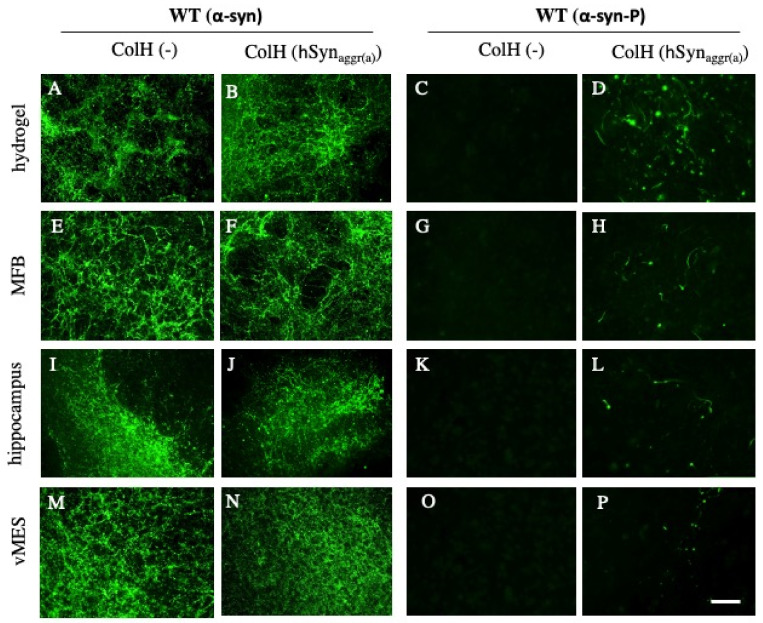
*Immunostainings for alpha synuclein (α-syn) and phosphorylated α-syn-P in slices taken from**wild-type (WT) mice.* Organotypic brain slices were prepared from WT mice and, after 1 week, a control-empty collagen hydrogel (ColH(-)) (**A**,**C**,**E**,**G**,**I**,**K**,**M**,**O**) or a collagen hydrogel loaded with aggregated active human α-syn (ColH(α-syn_agg_)) (**B**,**D**,**F**,**H**,**J**,**L**,**N**,**P**) was placed on the slices. The slices were fixed in culture after 8 weeks and stained for α-syn or phosphorylated α-syn (α-syn-P). Immunostainings were evaluated in the area of the hydrogel (**A**–**D**), in the medial forebrain bundle area (MFB; **E**–**H**), in the hippocampus (**I**–**L**) and in the ventral mesencephalon (vMES; **M**–**P**). A strong α-syn immunostaining was seen when an empty collagen hydrogel (**A**,**E**,**I**,**M**) or a collagen hydrogel loaded with α-syn (**B**,**F**,**J**,**N**) was placed on WT slices, pointing to endogenous mouse α-syn expression. Immunostainings for α-syn-P showed no staining after application of control collagen hydrogels (**C**,**G**,**K**,**O**), but WT slices loaded with aggregated active α-syn exhibited strongly stained elongated structures or round aggregates in all the brain regions evaluated (**D**,**H**,**L**,**P**). Scale bar, *p* = 75 µm (**A**,**B**,**E**,**F**,**I**,**J**,**M**,**N**) and 25 µm (**C**,**D**,**G**,**H**,**K**,**L**,**O**,**P**).

**Figure 7 biomolecules-12-00163-f007:**
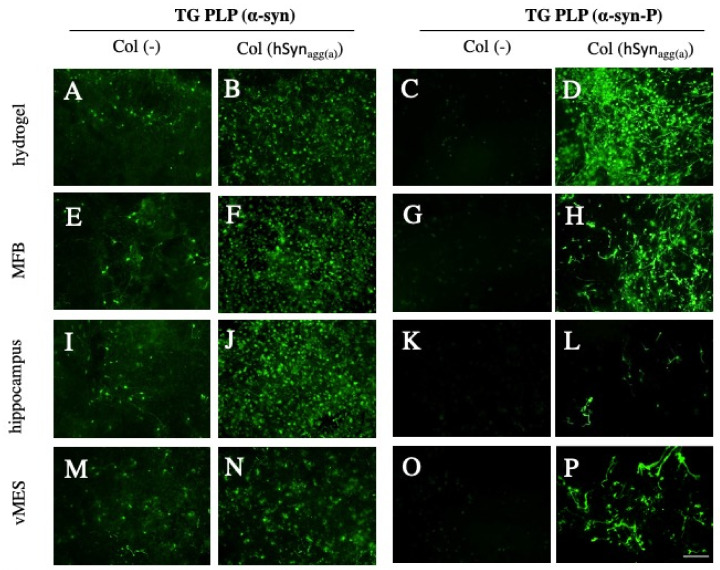
*Immunostainings for alphα-synuclein (α-syn) and phosphorylated α-syn-P in slices taken from transgenic (TG) PLP mice.* Organotypic brain slices were prepared from TG PLP mice and, after 1 week, a control-empty collagen hydrogel (ColH(-)) (**A**,**C**,**E**,**G**,**I**,**K**,**M**,**O**) or a collagen hydrogel loaded with aggregated active human α-syn (ColH(α-syn_agg_)) (**B**,**D**,**F**,**H**,**J**,**L**,**N**,**P**) was placed on the slices. The slices were fixed in culture after 8 weeks and stained for α-syn or phosphorylated α-syn (α-syn-P). Immunostainings were evaluated in the area of the hydrogel (**A**–**D**), in the medial forebrain bundle area (MFB; **E**–**H**), in the hippocampus (**I**–**L**) and in the ventral mesencephalon (vMES; **M**–**P**). A very low α-syn immunostaining was seen when an empty collagen hydrogel (**A**,**E**,**I**,**M**) was placed on transgenic slices. Application of a collagen hydrogel loaded with α-syn (**B**,**F**,**J**,**N**) markedly enhanced staining in TG mice, pointing to stronger endogenous mouse α-syn expression. Immunostainings for α-syn-P showed no staining for control collagen hydrogels (**C**,**G**,**K**,**O**), but transgenic slices loaded with aggregated active α-syn exhibited very strongly stained elongated structures or round aggregates in all the brain regions evaluated (**D**,**H**,**L**,**P**). Scale bar, *p* = 75 µm (**A**,**B**,**E**,**F**,**I**,**J**,**M**,**N**) and 150 µm (**C**,**D**,**G**,**H**,**K**,**L**,**O**,**P**).

**Figure 8 biomolecules-12-00163-f008:**
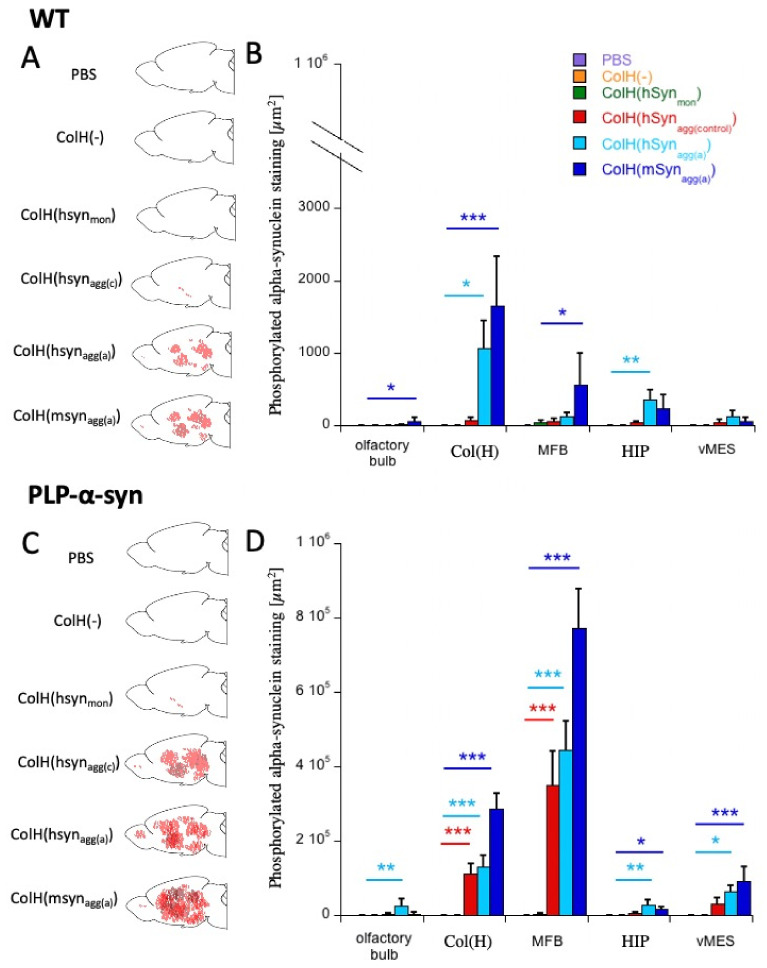
*Quantification of alphα-synuclein (α-syn) spreading in organotypic brain slices.* Organotypic brain slices were prepared from wild-type (WT; **A**,**B**) or transgenic PLP α-syn (**C**,**D**) mice. After 1 week in culture with phosphate-buffered saline (PBS), an empty control collagen hydrogel (ColH(-)), or collagen hydrogels loaded with α-syn monomers (ColH(hsyn_mon_)) or aggregates (ColH(hsyn_agg(control)_), ColH(hsyn_agg(a)_) and ColH(msyn_agg(a)_) were placed on the slices. After 8 weeks in culture, the slices were fixed and stained for α-syn-P. The total area occupied by α-syn aggregates was quantified with ImageJ. (**A**,**C**) give a schematic representation of α-syn-P immunoreactivity in the sagittal brain sections. (**B**,**D**) show the computer-assisted quantification of α-syn staining. Values are given as mean ± SEM optical density. The statistical analyses were performed by a one-way ANOVA with a subsequent Fisher LSD post hoc test (* *p* < 0.05; ** *p* < 0.001; *** *p* < 0.001). Abbreviations: MFB, medial forebrain bundle; HIP, hippocampus; vMES, ventral mesencephalon; Col(H), collagen hydrogel.

**Figure 9 biomolecules-12-00163-f009:**
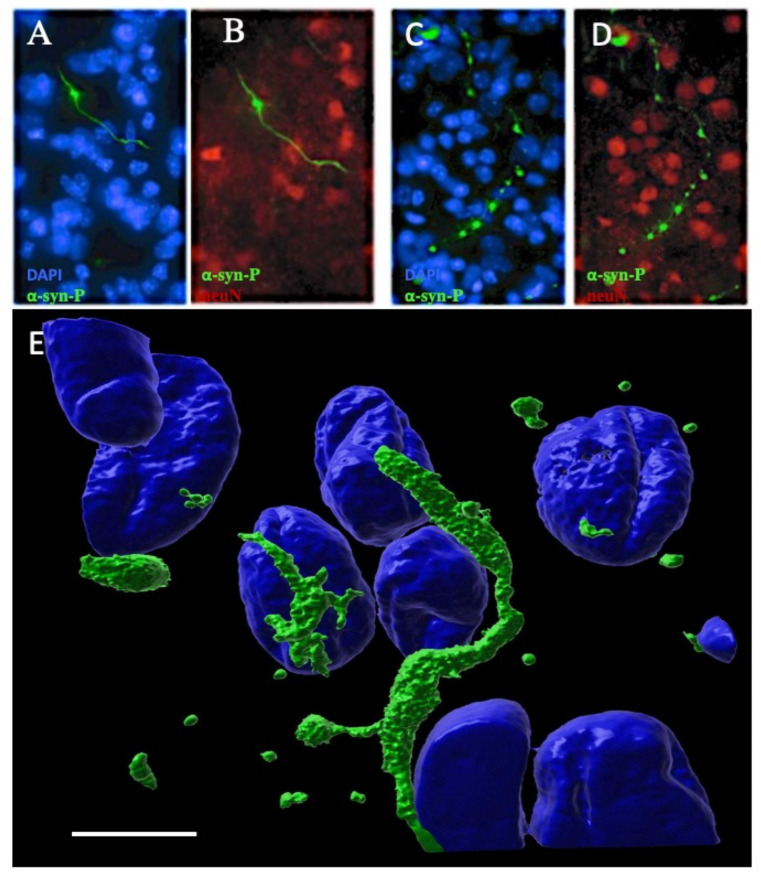
*Close-up images of alphα-synuclein (α-syn) aggregates.* Organotypic brain slices of wild-type mice were prepared, collagen hydrogels with ColH(hsyn_agg(a)_) were applied and the slices were incubated for 8 weeks. The slices were then fixed and stained for α-syn-P (green; Alexa-488) and nuclear DAPI (blue) or neuronal neuN (red; Alexa-546). Note that long fibrillar aggregates (**A**,**B**) or discontinued lines formed by dot-like aggregates (**C**,**D**) were observed. Reconstructed 3D depiction of confocal images showed that these aggregates surrounded the nuclei and were only some µm in size (**E**). Scale bar, E = 60 µm (**A**–**D**) and 7 µm (**E**).

**Figure 10 biomolecules-12-00163-f010:**
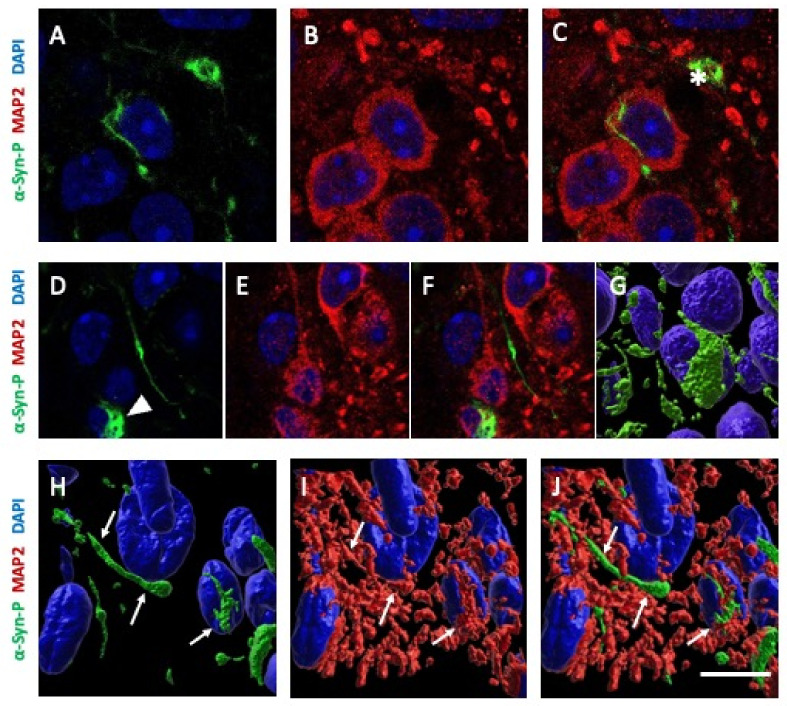
*Co-localization of alphα-synuclein (α-syn) aggregates in neuronal processes.* Organotypic brain slices of wild type mice were prepared, collagen hydrogels with ColH(hsyn_agg(a)_) were loaded and the slices were incubated for 8 weeks. The slices were then fixed and stained for α-syn-P (green; Alexa-488) and nuclear DAPI (blue fluorescent) or neuronal microtubuli-associated protein-2 (MAP2; red; Alexa-546). Aggregates with small fibrillary appearance were observed in proximity to the nuclei of neurons (**A**–**C**). A white asterisk indicates a degenerating neuron with α-syn pathology (**C**). In some neurons, α-syn aggregates noticeably surrounded the neuronal nuclei (**D**–**G**; white arrowhead in **D**). (**G**) depicts the nucleus pointed at by a white arrowhead in a 3D reconstructed image, exhibiting the nucleus wrapped into α-syn aggregates. Note the degenerated axon with high α-syn load. Some α-syn aggregates co-localized with MAP2 (white arrows) depicted in 3D reconstructed images (**H**–**J**). Scale bar, J = 15 µm (**A**–**C**), 12 µm (**D**–**F**), 6 µm (**G**) and 8 µm (**H**–**J**).

**Figure 11 biomolecules-12-00163-f011:**
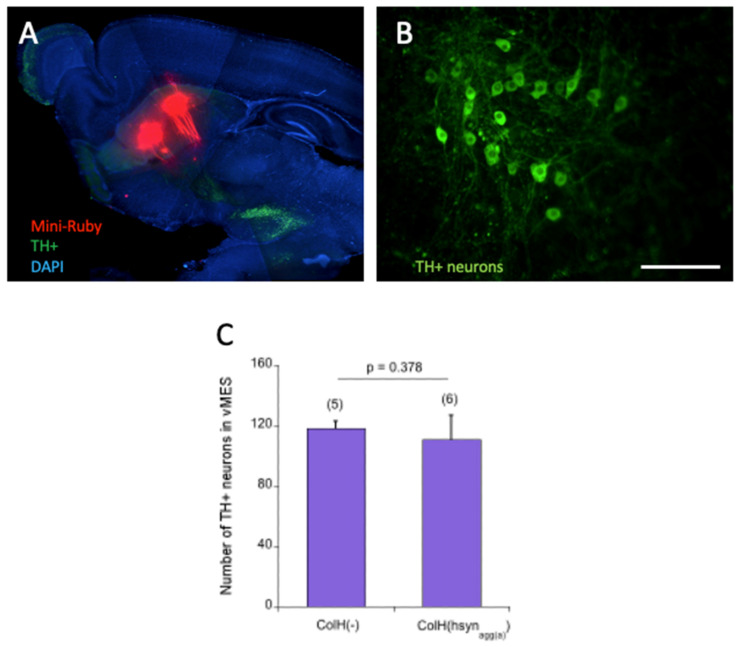
*Effect of**alphα-synuclein (α-syn) on dopaminergic neurons.* (**A**) Sagittal organotypic brain slices were cultured for 1 week; then, Mini-Ruby crystals were applied onto the striatum and retrograde transport was evaluated after 2 h. Note that Mini-Ruby (red fluorescent) stained the striato-nigral pathway but did not reach the dopaminergic neurons (stained by tyrosine-hydroxylase, TH; green fluorescent). DAPI was used to counterstain the slices. (**B**) Dopaminergic TH+ neurons survived culturing for 9 weeks when incubated with glial-cell-line-derived neurotrophic factor. (**C**) The quantitative effects of collagen-loaded α-syn (ColH(α-syn)) versus a collagen negative control (ColH(-)) on the number of dopaminergic neurons in the vMES are shown. Values are given as mean ± SEM, n gives the number of animals; the statistical analyses were performed by a one-way ANOVA with a subsequent Fisher LSD post hoc test. Scale bar, B = 900 µm (**A**) and 120 µm (**B**).

## Data Availability

The data that support the findings of this study are available on request from the corresponding author.

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
