# Peer review of "Spreading of Aggregated α-Synuclein in Sagittal Organotypic Mouse Brain Slices"

_biomolecules, 2022, doi:10.3390/biom12020163_

Round 1

Reviewer 1 Report

Uçar et al presented an interesting article on Spreading of aggregated α-synuclein in sagittal organotypic mouse brain slices.

The paper contains large results, although this section needs a lot of refinement. While there are not that many mistakes in the text - sometimes double space, or wildtype spelled together, but it's hard to refer to a particular passage because there are no line markings. I'm not sure if this is already the case in Biomolecules or if the authors didn't follow the scheme.

Figures are usually described twice, apart from the caption there is an entry such as Figure1 next to or below, except for Figures 4 and 5.
The font used is probably different, e.g. in Fig.1 panel E total pixelosis. 
The order is swapped - how come in the manuscript it is Fig8 first and then 7?
Fig.11 shows absolutely nothing, the quality is awful
I also don't understand the analysis - was fluorescence intensity analysis done in case of Fig.6?
What about densitometry of Westernblotting results, I don't see any analysis for Fig 1 or 3? Although the results section has potential, it was prepared poorly 

A big plus for the Limitations of the Study subsection. It is good to see that the authors themselves recognize the problems. Although here the question arises: what other models are known? Maybe some form of table comparing advantages and disadvantages would be interesting. 
as others showed that organotypic brain slice cultures from postnatal transgenic mice display a pathology similar as seen in Alzheimer's
disease.---- no citation, it would be useful to expand the thread here

The authors mention that they created a similar model for beta amyloid -- is anything further being done in this specific direction? Are you working on other aggregates? 

I am wondering what the authors were most driven by? what is the goal, the mechanism to confirm the prion-like synuclein distribution theory I understand but what next?

Author Response

Ad Referee 1

Uçar et al presented an interesting article on Spreading of aggregated α-synuclein in sagittal organotypic mouse brain slices.

The paper contains large results, although this section needs a lot of refinement. While there are not that many mistakes in the text - sometimes double space, or wildtype spelled together, but it's hard to refer to a particular passage because there are no line markings. I'm not sure if this is already the case in Biomolecules or if the authors didn't follow the scheme.

Response: We apologize. We have used the layout of the Journal. The line markings have been added now. Double spaces, style mismatch and mistakes are fixed.

Figures are usually described twice, apart from the caption there is an entry such as Figure1 next to or below, except for Figures 4 and 5.

Response: This has been corrected.

The font used is probably different, e.g. in Fig.1 panel E total pixelosis. 

Response: This has been corrected. Fig1E has been replaced.

The order is swapped - how come in the manuscript it is Fig8 first and then 7?

Response: We are sorry for this confusion. This has been corrected.

Fig.11 shows absolutely nothing, the quality is awful

Response: We are sorry, but the transfer of this image did not work properly. This has been corrected. The Figure 11C has been replaced.

I also don't understand the analysis - was fluorescence intensity analysis done in case of Fig.6?

Response: We did not quantify the intensity of a-syn stainings that is shown in the first 2 columns of Fig.6, as it shows the total a-syn. The analysis shown in Figure 8 is the quantification of the area stained by antibodies against phosphorylated a-syn-P (last 2 columns of Fig.6), which is used as an indication of a-syn aggregates. The same is valid for the Fig.7. The Western Blots only show the verification of the used proteins and aggregates and give a non-quantitative overview of changes over time.

What about densitometry of Westernblotting results, I don't see any analysis for Fig 1 or 3? Although the results section has potential, it was prepared poorly 

Response: Densiometry analysis was not carried out for the Western blots in Fig 1 and 3, as the n numbers were too low (n=2 in Fig 1 per group and n=1 in Fig 3). It was not the focus to study the time course, but rather to focus on the 8-week incubation time, similary as shown for the beta-amyloid story (see Moelgg et al., 2021).

A big plus for the Limitations of the Study subsection. It is good to see that the authors themselves recognize the problems. Although here the question arises: what other models are known? Maybe some form of table comparing advantages and disadvantages would be interesting. as others showed that organotypic brain slice cultures from postnatal transgenic mice display a pathology similar as seen in Alzheimer's disease.---- no citation, it would be useful to expand the thread here

Response: There are various published models of a-syn aggregation or the related synucleinopathies, ranging from primary cell cultures to in vivo models. The advantages/disadvantages of using organotypic brain slice cultures are well known and published in many previous research papers and reviews (see Humpel, 2015), which is mentioned in the Introduction section. Additionally, previous models of a-syn spreading with organotypic brain slice cultures are summarized in the Introduction section. The advantage of our model compared to these models is the use of a sagittal whole brain section in our model that allows to observe different brain regions together, while the previous models are comprised of a single brain region (ex. hippocampal slices). This advantage was also stated in the Introduction and Discussion sections. Making a table of previous models is out of scope of this study, and would rather be useful in a review/perspective, however, the sections about comparison of the models has been extended in the MS (line 782).

The authors mention that they created a similar model for beta amyloid -- is anything further being done in this specific direction? Are you working on other aggregates? 

Response: The work on beta amyloid model has been published and also discussed in detail especially regarding other issues (see Moelgg et al., 2021). Currently there is another study running in our group for modeling the spreading of tau tangles/aggregates (Korde and Humpel, in preparation).

I am wondering what the authors were most driven by? what is the goal, the mechanism to confirm the prion-like synuclein distribution theory I understand but what next?

Response: The main goal of our study was to create a model of a-syn spreading with a whole sagittal brain slice to include regional differences in the pathology. This characterized model later can find applications in investigating the neurotoxic mechanisms of synucleinopathies and for drug screening. This has been better focused and described now.

Reviewer 2 Report

The paper by Ucar et al. it is an interesting manuscript focused on a-Syn spreading in brain slices. Given the relevance of a-Syn dynamics in diseases such as PD or MSA this study can be of relevance for the community. The paper is well written and the methods are well described. 

Before the paper can be considered for publication the following points are to be addressed: 

Main comments: 

  • section 4 (Discussion) is currently divided in section and looks like a repetition of the results and introduction section. This need to be amended to facilitate the reader in following the experimental design.
  • Section 5 (Conclusions) consists in just a few words and it is unclear to me if this paper is presented as a methodological advance or an investigation on aSyn spreading. The authors should expand on this. 
  • Discussion, section 4.9. If the authors think the lack of aSyn oligomers in the model is responsible for the missing protein toxicity, this should be discussed clearly as this is still a key debate point for the community.  

Other comments:

  • In the abstract section the first sentence states clearly that aSyn has prion like properties; at the same time, in the introduction is stated that this is an hypothesis under investigation. This needs to be clarified.
  •  figure 5: the authors state that no mRNA change is observed, however a trend appears in the plot. if this is not statistically significant, p values should be reported in the legend and in the MS text
  • in section 3.6 figure 8 is discussed before figure 7, which is discussed in section 3.7. The 2 figures appear also in the inverted order in the MS. The text and figures order needs to be amended for consistency. 
  • Figure 9: please add clear legend for image E
  • Figure 11: panel C is blurred and difficult to see
  • in results, section 4.3 please correct typo (remove extra dot)

Author Response

Ad Referee 2

The paper by Ucar et al. it is an interesting manuscript focused on a-Syn spreading in brain slices. Given the relevance of a-Syn dynamics in diseases such as PD or MSA this study can be of relevance for the community. The paper is well written and the methods are well described. Before the paper can be considered for publication the following points are to be addressed: 

section 4 (Discussion) is currently divided in section and looks like a repetition of the results and introduction section. This need to be amended to facilitate the reader in following the experimental design.

Response: I am really sorry that the referee thinks that. In fact, we think that this is a typical way to discuss any issue of the experimental design. This has also been done in our previous work in Biomolecules (Moelgg et al., 2021). I know that also there are ways to fuse the Results and Discussion into one part, which I do not like so much, and is also not the style of the Journal. We would like to keep it this way, or did I missunderstand this critics?

Section 5 (Conclusions) consists in just a few words and it is unclear to me if this paper is presented as a methodological advance or an investigation on aSyn spreading. The authors should expand on this. 

Response: This has been better focused and expanded.

Discussion, section 4.9. If the authors think the lack of aSyn oligomers in the model is responsible for the missing protein toxicity, this should be discussed clearly as this is still a key debate point for the community.  

Response: In the PLP-a-syn transgenic mouse, a-syn oligomers are generated at a later stage of life, i.e. 6 months of age, at the time of nigral neurodegeneration (see Refolo et al. 2018). Therefore, it is considered that a-syn oligomers play a pivotal role in the neurodegeneration in vivo. This has been better discussed (lines 735, 631).

In the abstract section the first sentence states clearly that aSyn has prion like properties; at the same time, in the introduction is stated that this is an hypothesis under investigation. This needs to be clarified.

Response: The abstract has been adapted accordingly.

 figure 5: the authors state that no mRNA change is observed, however a trend appears in the plot. if this is not statistically significant, p values should be reported in the legend and in the MS text

Response: Yes we agree, the p-values are added (Fig 5 line 418 and line 275).

in section 3.6 figure 8 is discussed before figure 7, which is discussed in section 3.7. The 2 figures appear also in the inverted order in the MS. The text and figures order needs to be amended for consistency. 

Response: This has been corrected, thank you.

Figure 9: please add clear legend for image E

Response: We are sorry, this has been added.

Figure 11: panel C is blurred and difficult to see

Response: We are very sorry for this very bad Image. This was a failure of the image transfer. This has been corrected and replaced.

in results, section 4.3 please correct typo (remove extra dot)

Response: This has been corrected. Thank you.

Round 2

Reviewer 1 Report

Authors adressed all concerns